# Dye Plants Derived Carbon Dots for Flexible Secure Printing

**DOI:** 10.3390/nano12183168

**Published:** 2022-09-13

**Authors:** Linlin Li, Yuanyuan Han, Lihua Wang, Wei Jiang, Haiguang Zhao

**Affiliations:** 1College of Textiles & Clothing, Qingdao University, No. 308 Ningxia Road, Qingdao 266071, China; 2State Key Laboratory of Bio-Fibers and Eco-Textiles, College of Physics, Qingdao University, No. 308 Ningxia Road, Qingdao 266071, China

**Keywords:** dye plants, carbon dots, efficient, environmental friendliness, flexible secure printing

## Abstract

Carbon dots (C-dots) are fluorescent nanomaterials, exhibiting excellent structure-dependent optical properties for various types of optical and electrical applications. Although many precursors were used for C-dots production, it is still a challenge to produce high-quality C-dots using environmentally-friendly natural precursors. In this work, multiple-colored colloidal C-dots were synthesized via a heating reaction using natural plant dyes as precursors, for example, Indigo, *Carcuma longa*, and *Sophora japonica* L. The as-prepared C-dots have absorption in the UV range of 220 to 450 nm with the typical emission ranging from 350 to 600 nm. The as-obtained C-dots have a quantum yield as high as 3.8% in an aqueous solution. As a proof-of-concept, we used the as-prepared C-dots as fluorescence inks for textile secure printing. The printed patterns are almost invisible under daylight and have distinct and clear patterns under 365 and 395 nm light, proving the great potential in optical anti-counterfeiting. This work demonstrates the advanced strategy for high-performance C-dots production from natural dyes and their potential application in flexible secure printing systems.

## 1. Introduction

Carbon dots (C-dots) are small-sized zero-dimensional nanomaterials with a typical size of less than 10 nm. They typically consist of earth-abundant C, N, and O elements. Owing to their excellent optical and electrical properties, C-dots have gained tremendous attention in recent years [1,2,3]. For example, most of the C-dots have low toxicity, favorable biocompatibility, excellent photostability, and good water solubility. Nowadays, C-dots have been employed frequently as building blocks for a variety of applications, including optical sensors, light-emitting diodes, bio-imaging, photocatalysis, and nanomedicine [4,5,6,7,8,9]. Very recently, the C-dots have been used for fluorescent anti-counterfeiting technology [10]. Compared to commonly used anti-counterfeiting materials (e.g., quantum dot, dyes/polymer, upconversion nanoparticles), C-dots have very good water solubility and they are stable after being printed on the substrate in harsh conditions (e.g., high humidity) [11,12,13]. In addition, they can be produced via simple chemical approaches [1]. For instance, Zhu et al. [14] used citric acid and ethylenediamine as raw materials to synthesize C-dots by hydrothermal carbonization and they further used the as-prepared C-dots for the anti-counterfeiting system. Tian et al. [15] used nitrilotriacetic acid (NTA) as the carbon source via solvothermal reaction to obtain N-doped C-dots, which exhibited clear fluorescent patterns after printing. Zhou et al. [16] used Ti_3_C_2_T_x_ MXene as the carbon source and nitrogen doping source for C-dots synthesis and further used it for fluorescent inks. These studies demonstrated the applicability of C-dots as fluorescent materials for secure printing.

Natural dyes are abundant natural resources that can be extracted from minerals, plants, or animals [17]. Natural dyes have attracted a lot of attention due to their high qualities, such as being green and safe, antibacterial and anti-inflammatory, low-cost, and environmental compatibility [18,19]. In addition, natural dyes are rich in C, O, and other elements, which are suitable raw materials for the production of C-dots. At present, research on natural dyes is mainly focused on fiber dyeing and dye-sensitized solar cells [18,20,21,22], and to the best of our knowledge, there is still little study on synthesizing C-dots using natural dyes. The use of dyes obtained from nature products for preparing C-dots for anti-counterfeiting codes not only provides a new approach in the field of secure printing, but also presents great significance for the environment.

In this work, we demonstrate the synthesis of colloidal C-dots using natural dyes as raw materials and used the as-synthesized C-dots for the anticounterfeiting system. Three natural dyes, including Indigo, *Carcuma longa*, and *Sophora japonica* L., were used as raw materials to prepare C-dots via a heating reaction. The prepared C-dots have fluorescence lifetimes of 1.0 to 3.9 ns and a quantum yield (QY) of as high as 3.8% in an aqueous solution. Using the prepared C-dots as fluorescent inks, after printing, we can obtain clear anti-counterfeiting patterns, which show a great development of flexible secure printing systems using low-cost and non-toxic natural dyes.

## 2. Materials and Methods

### 2.1. Materials

The natural plant dyes were obtained from Changzhou Meisheng Biomaterials Co., Ltd., Changzhou, China. Sinopharm Chemical Reagent Co. (Shanghai, China) provided the chemicals that were used in the studies, including sodium polyacrylate, ethylene glycol, fatty alcohol ethoxylates, and anhydrous ethanol. All chemicals are reagent grade and have been used directly without additional purification.

### 2.2. Synthesis of C-Dots

The C-dots were synthesized via heating approach. Natural plant dyes containing Indigo dyes, *Carcuma longa* deys, and *Sophora japonica* L. dyes were dissolved separately in anhydrous ethanol or water, with a final concentration of 5 mmol/L, then the mixture was dissolved by ultrasonic shaking for 20 min. Centrifuging the as-obtained mixture for five minutes at a speed of 6000 r.p.m. can remove the insoluble materials. Subsequently, the supernatant was poured into a Teflon-lined autoclave and heated for 6 h at 180 °C. Finally, the purification was carried out using a dialysis bag (3500 Da) for 12 h and the dialysate was changed every 3 h. The resulting solution was applied to the preparation of ink or optical characterization.

### 2.3. Ink Preparation and Printing

The ink was produced using the as-prepared C-dots. Ink formulation consists of primary alcohol ethoxylate, polyacrylate, ethylene glycol, C-dots, and water in the proportions of 1%, 0.005%, 1%, 0.35%, and 97.64%, respectively. All materials were mixed and stirred on a magnetic mixer (800 r.p.m.) for 4 h to achieve homogeneous mixing. Subsequently, mixed liquids were filtered through using 0.22 μm filters. The final ink was obtained at a C-dots concentration of 0.0035 mg/mL.

### 2.4. Characterizations

The C-dots were examined by transmission electron microscope (TEM) using a JEOL 2100F TEM (Tokyo, Japan) with a magnification of 20 nm to observe the microscopic morphology of the C-dots; the lattice was examined using a high resolution TEM (HRTEM) with a magnification of 5 nm. A UV-Vis spectrometer with a scan range of 200–600 nm, the Lambda 750, was used to measure the UV-Vis absorption spectra of the C-dots. The fluorescence lifetime and quantum yield (QY) of the C-dots were tested using a steady-state transient fluorescence spectrometer and an absolute quantum efficiency tester. An Edinburgh FLS1000 equipment (Livingston, UK) was used to characterize the steady-state photoluminescence (PL) spectra of C-dots. The Fourier transform infrared (FT-IR) was performed on a Nicolet 6700 FT-IR (Thermo Fisher Scientific, Waltham, MA, USA)spectrometer with 32 scans. X-ray photoelectron spectroscopy (XPS) was used to characterize the elements and functional groups on the surface of the C-dots, testing the full spectrum and the fine spectrum of each element. Zeta potential characterization was obtained using the Zeta Sizer Nano-ZS (Malvern Instrument, Inc., London, UK).

## 3. Results and Discussion

### 3.1. Synthesis and Structure of C-dots

Figure 1 illustrates the straightforward one-step heating process by putting the precursor and solvent in an autoclave that was used to produce the C-dots. Three different natural plant dyes (Indigo, *Carcuma longa*, and *Sophora japonica* L.) were used as precursors, and ethanol was used as a solvent (Figure 1a–c). The details of C-dots preparation were included in the Experimental Section. Specifically, all the reacted natural plant dyes were dissolved in ethanol at a concentration of 5 mmol/L and reacted under high pressure at 180 °C for 6 h. In order to investigate the effect of different solvents, we also used water as a solvent for the reaction. As most of the natural dyes were extremely insoluble in water, the concentration of the prepared C-dots was very low and the optical properties are shown in Appendix A. From the point of view of improving the reaction yield of the C-dots, we finally used ethanol as the solvent for the preparation of the C-dots.

TEM and HR-TEM were used to examine the morphologies of the three different types of C-dots, and the results are displayed in Figure 2 and Appendix A. As shown in Figure 2, three types of C-dots have quasi-spherical shape. With an average diameter of 3.5 ± 0.5 nm (Indigo), 2.6 ± 0.5 nm (*Carcuma longa*), and 5.2 ± 0.7 nm (*Sophora japonica* L.), respectively, they had a limited size distribution (Appendix A). Three types of the C-dots have clear lattice spacing of 0.22, 0.21, and 0.28 nm, respectively (Figure 2d–f), which correspond to the lattice planes of graphene [23].

To better characterize the chemical composition of the C-dots prepared with these natural dyes, the Fourier transform infrared (FT-IR) and X-ray photoelectron spectroscopy (XPS) measurements were carried on for the C-dots. The natural structure of the natural dyes used in this work is shown in Table 1. Typically, Indigo dyes are trans-symmetrical bisindole structures with the molecular formula C_16_H_10_N_2_O_2_ [24], *Carcuma longa* dyes are predominantly β-diketones with a trans-geometric structure with the molecular formula C_21_H_2_O_6_ [25], and *Sophora japonica* L. dyes are mainly composed of rutinosides (rutin) which are flavanol ligands with the molecular formula C_27_H_30_O_16_ [26]. As seen by Figure 3a, in comparison with the characteristic peaks of the Indigo dyes, C-dots produced using Indigo dyes have the typical peak located at 3100–3700 cm^−1^, 1718 cm^−1^, and 1410 cm^−1^, which can be assigned to the O–H/N–H, C=C/C=O/N–H, and C–O/C–N stretching vibrations, respectively. The C–H/C–C and C–C stretching vibrations are attributed to the weak peaks around 2900 cm^−1^ and 1500 cm^−1^, respectively [27,28,29]. A similar phenomenon was found for the C-dots produced using *Carcuma longa* and *Sophora japonica* L. as precursors [27,28,29]. While the C–H/C–C and C–C characteristic peaks near 2900 cm^−1^ and 1500 cm^−1^ almost disappear for the C-dots produced by *Carcuma longa* and *Sophora japonica* L.

XPS measurements were used to further characterize the chemical composition of these C-dots (Figure 3 and Appendix A). The C-dots produced using Indigo have C 1 s at 285 eV, O 1 s at 533 eV, and N 1 s at 401 eV, with elemental contents of 71.98%, 17.24%, and 3.89%, respectively (Appendix A). Due to the impurities of the natural dyes, we also found other elements in the XPS spectra of as-prepared C-dots. The high-resolution spectra of the C 1 s show three distinct peaks (C–C at 284.8 eV, C–O/C–N at 286.2 eV, and C=O at 288.6 eV) (Figure 2d).

The full XPS spectra of the C-dots derived from *Carcuma longa* contain the elements of C, O, N, Cl, Si, and S with contents of 78.54%, 17.21%, 1.69%, 0.46%, 1.61%, and 0.49%, respectively (Appendix A). The content of elements other than C and O is due to the impurities in the dyes. According to Figure 3e, the three peaks of the C 1 s peaks of the C-dots derived from *Carcuma longa* are the C–C peak at 284.8 eV, the C–O peak at 285.9 eV, and the C=O group at 288.7 eV. Furthermore, the XPS spectra of C-dots derived from *Sophora japonica* L. have similar signals, which have C 1 s at 285 eV, O 1 s at 533 eV with elemental contents of 64.47%, and 26.56%, respectively. Other elements (N, Cl, and Se) are dyes’ impurities. The C 1 s peaks of the C-dots derived from *Sophora japonica* L. show three distinct peaks that C–C at 284.8 eV, the C–O at 286.2 eV, and the C=O group at 288.4 eV, as shown in Figure 3f.

### 3.2. Optical Properties of C-Dots

The as-prepared C-dots dispersed in water have a light yellow color under room light. Upon 395 nm illumination, the C-dots aqueous solutions show blue (Indigo), cyan (*Carcuma longa*), and bluish green (*Sophora japonica* L.) colors (inset of Figure 1d–f). We also thoroughly examined the optical characteristics of C-dots made from natural dyes, such as their UV-Vis absorption, PL spectra, and transient PL spectra. All of the C-dots exhibit absorption between 220–450 nm, as shown in Figure 4a–c. All C-dots have excitation-dependent PL behaviors. The reason for this phenomenon is inferred to be the presence of surface defects such as oxygen-containing functional groups or nitrogen-containing functional groups in the surface structure of the C-dots [28].

Moreover, the PL spectra of the natural dyes were measured. Only the *Carcuma longa* dyes have very bright yellow emission. As shown in Figure 4d, *Carcuma longa* dyes have an excitation-independent PL behavior. This behavior is different from the C-dots derived from the natural dyes, indicating that we can successfully prepare the C-dots from *Carcuma longa* dyes. The surface defects are typically the cause of the excitation-dependent PL behavior [30], resulting in multiple-energy states in the C-dots, other than the stable single interband.

A schematic diagram of the energy band structure of the *Carcuma longa* dyes was shown in Figure 4f. The *Carcuma longa* dyes have a dominant luminescence center, which contributes to the excitation-independent photoluminescence properties. In contrast, in the C-dots represented in the right panel of Figure 4f, there are multiple recombination centers, explaining that the C-dots exhibit excitation-dependent PL behavior.

The QYs were 3.8%, 0.8%, and 1.0% for the C-dots synthesized using Indigo, *Carcuma longa*, and *Sophora japonica* L., respectively (Table 1). The QY of as-prepared C-dots derived from Indigo is at a relatively high level compared to the 0.76% QY of C-dots obtained by Zhu et al. [31], which also produced the C-dots using Indigo dyes. Compared to other natural materials, the QY of C-dots obtained by Hsu et al. [32] using coffee grounds was 3.8%. Soy milk was utilized by Zhu et al. [33] to synthesize C-dots with a 2.6% QY. The lower QYs of the C-dots derived from *Carcuma longa* and *Sophora japonica* L. may be related to the fact that the dye composition is mainly C, O, and contains fewer functional groups of other elements [34].

The lifetimes of the C-dots were measured using transient PL spectra (Figure 4e) These decay curves were all biexponential fits (τ1 and τ2). The three decay curves are all found to have a fast component (1: 1.6, 0.3, 0.9 ns) and a slow component (2: 8.7, 5.2, 4.5 ns), according to the fitted parameters (as shown in Table 1). The average lifetimes for C-dots produced using Indigo, *Carcuma longa*, and *Sophora japonica* L. were 3.0 ns, 3.9 ns, and 1.0 ns, respectively.

### 3.3. Optical Anti-Counterfeiting System

Benefiting from the hydrophilic functional groups on their C-dots surface, the as-prepared C-dots can be dispersed in an aqueous solution very well. The carbon point can be almost completely dissolved in water at room temperature under stirring in a magnetic mixer. Some printing-friendly additives can be added such as primary alcohol ethoxylate, polyacrylate, and ethylene glycol to the C-dots solution then mixed well to produce the printing ink. The fluorescence performance of the inks at 365 nm and 395 nm are shown in Appendix A. Under different light irradiation, all three C-dots inks showed a red shift of the PL, which is consistent with the change in the optical properties of C-dots.

Fluorescent inks obtained from the preparation of C-dots with different luminescent colors can be used to achieve fluorescent anti-counterfeiting effects by printing patterns under UV light. The three C-dots in an aqueous solution were used as fluorescent inks in experiments using a digital inkjet printer (HFTX-P4290C). When C-dots inks are printed on non-fluorescent cotton fabric, the cotton fibers quickly recrystallize the inks. In the light, the C-dots ink-printed patterns are essentially undetectable, as seen in Figure 5. However, under 365 nm and 395 nm illumination, these printed patterns show vibrant fluorescence, and the lines of the pattern at 395 nm are more clearly defined. The fluorescent patterns vanished entirely when the stimulation was turned off. The patterns printed on cotton fabric using C-dots inks have very little blotting, which is very advantageous for achieving information encryption in daylight and information decryption in UV light. The patterns printed on cotton fabric using C-dots inks have very little blotting, which is very advantageous for achieving information encryption in daylight and information decryption in UV light. In addition, as seen in the pattern inset in Appendix A, the printed pattern is still clearly visible when the cotton fabric is folded. Utilizing the PL performance of C-dots made from plant dyes in response to UV light excitation has the benefits of being inexpensive while still providing a high level of security. After the printed image being stored for a month, the ink print pattern still has a clear pattern as shown in Appendix A. This result is benefited from the highly PL stability of the C-dots. As shown in Appendix A, the PL intensity of the C-dots dispersed in water maintains about 80% of its original value after 3-month storage (at 25 °C, humidity of 70%). Meanwhile we also tested the colloidal stability of the C-dots. As shown in Appendix A, the surface charge of the purified C-dots dispersed in water was measured by a Zeta Sizer Nano-ZS. The average value of the zeta potential of the C-dots derived from different natural dyes were −22–−28 mV, confirming that the C-dots have negative charge on the surface, contributed by the hydroxyl and carboxyl groups [35]. In view of the stability of the pattern, it is still important to coat a thin layer of hydrophobic polymer on the cotton, which can protect the pattern during the washing process. Meanwhile, it is possible to use the pattern for the label of the textiles and other applications without washing.

## 4. Conclusions and Perspectives

We have prepared fluorescent C-dots with good performance using three natural plant dyes (Indigo, *Carcuma longa*, and *Sophora japonica* L.) as only precursors. The lattice sizes of the three plant dyes C-dots are 3.5 ± 0.5 nm (Indigo), 2.6 ± 0.5 nm (*Carcuma longa*), and 5.2 ± 0.7 nm (*Sophora japonica* L.). The optical properties of the C-dots were tested and the three C-dots show significant absorption peaks at 220–450 nm and emission wavelengths at 350–600 nm. The emission wavelengths are red-shifted at different wavelengths of excitation, with a clear excitation-dependent optical behavior. The average fluorescence lifetimes are 3.0, 3.9, and 1.0 ns, and have a QY as high as 3.8% in an aqueous solution. As a proof of concept, the C-dots were configured as fluorescent inks and printed on cotton fabric. The printed patterns are almost invisible under daylight and have distinct and clear patterns under UV light, proving the great potential in optical anti-counterfeiting, and are also able to provide a reference for efficient anti-counterfeiting applications in environmental protection.

Considering the complexity of the natural dyes, future research still needs to focus on the composition effect on the optical properties of the C-dots. The surface passivation of the C-dots could be improved by surface post-treatment, which might enhance the QY of the C-dots.

## Figures and Tables

**Figure 1 nanomaterials-12-03168-f001:**
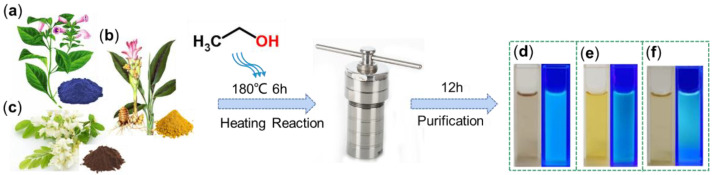
Schematic preparation of C-dots derived from natural plant dyes. (**a**–**c**) are the images of natural plants and dyes, corresponding to Indigo, *Carcuma longa*, and *Sophora japonica* L., respectively; (**d**–**f**) are C-dots solution upon room light (left) and 395 nm light illumination (right). The C-dots solutions were prepared via a heating reaction by using (**d**) Indigo, (**e**) *Carcuma longa*, and (**f**) *Sophora japonica* L. as precursors and ethanol as solvent.

**Figure 2 nanomaterials-12-03168-f002:**
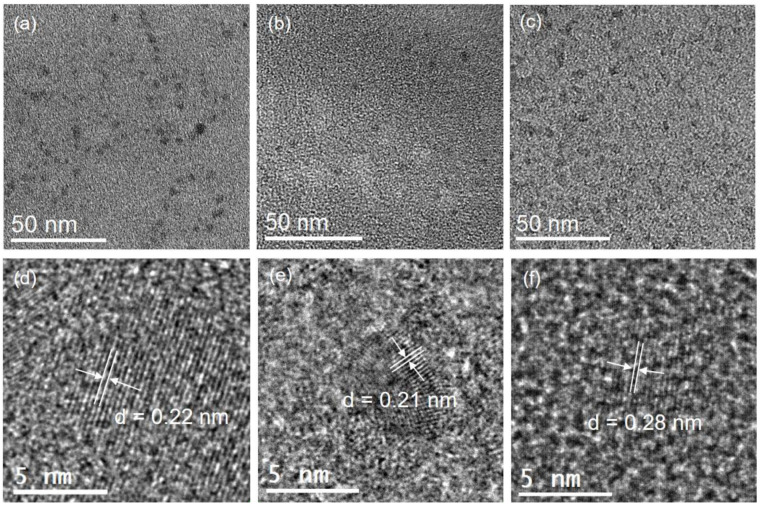
Typical TEM and HR-TEM images of the as-prepared C-dots. (**a**,**d**) C-dots derived from Indigo; (**b**,**e**) C-dots derived from *Carcuma longa*; (**c**,**f**) C-dots derived from *Sophora japonica* L.

**Figure 3 nanomaterials-12-03168-f003:**
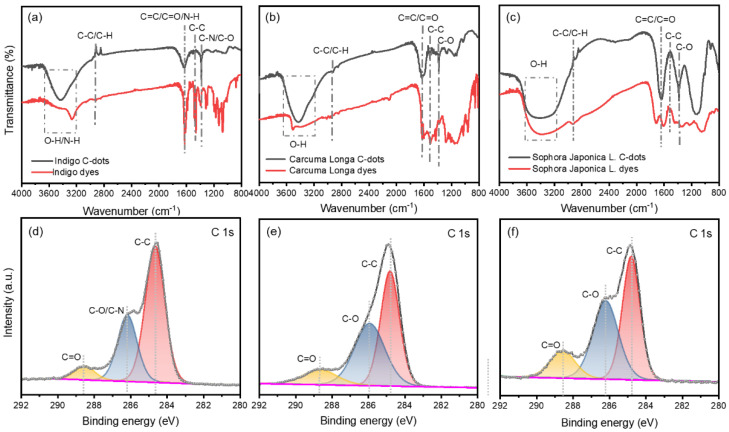
(**a**–**c**) are the FT-IR spectra of C-dots and dyes, corresponding to Indigo, *Carcuma longa*, and *Sophora japonica* L., respectively; (**d**–**f**) are high-resolution XPS C 1 s spectra corresponding to C-dots derived from Indigo (**d**), *Carcuma longa* (**e**), and *Sophora japonica* L. (**f**).

**Figure 4 nanomaterials-12-03168-f004:**
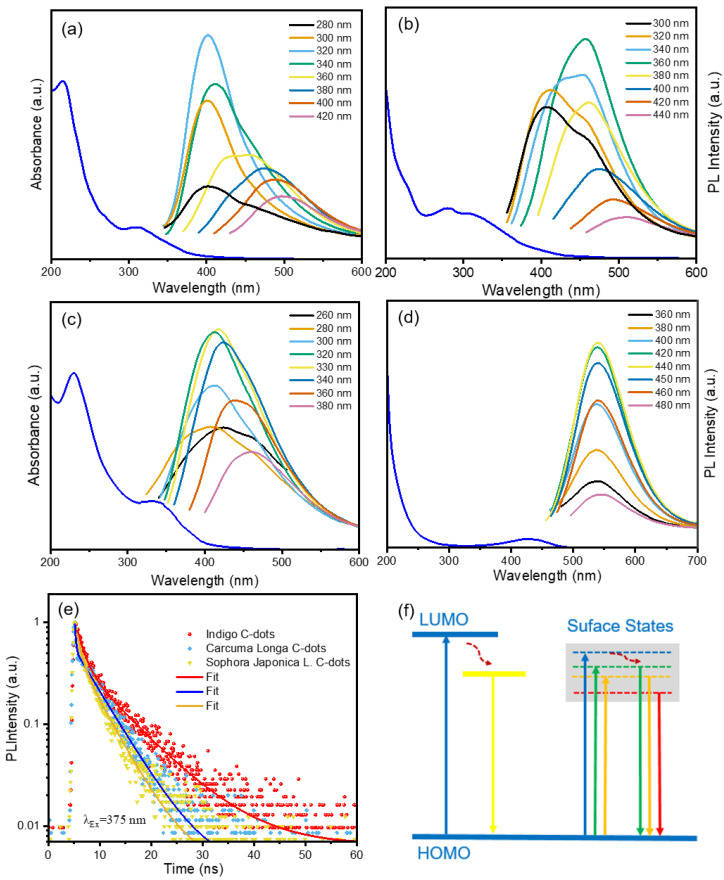
UV-Vis absorption and PL spectra of the C-dots dispersed in water using precursors of (**a**) Indigo, (**b**) *Carcuma longa*, and (**c**) *Sophora japonica* L.; (**d**) Absorption and PL spectra of the *Carcuma longa* dyes; (**e**) PL decay curves of the C-dots dispersed in water; (**f**) Possible electronic band structure for *Carcuma longa* dyes (left) and the C-dots (right).

**Figure 5 nanomaterials-12-03168-f005:**
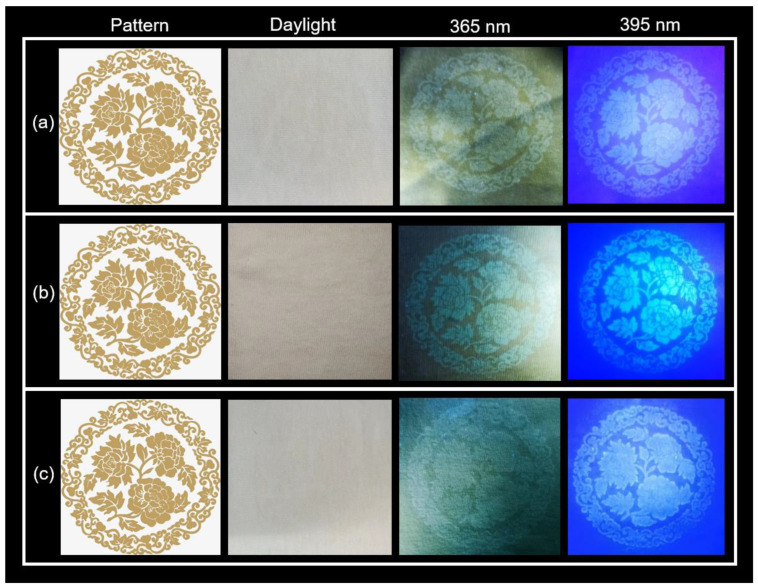
Anti-counterfeit patterns were designed and printed on cotton fabric under illumination at 365 nm and 395 nm. The C-dots were prepared using (**a**) Indigo, (**b**) *Carcuma longa*, and (**c**) *Sophora japonica* L.

**Table 1 nanomaterials-12-03168-t001:** The images of dyes and major components, absorption values, PL peaks, lifetimes, and QYs of the C-dots dispersed in water.

Raw Material	Major Component	Abs (nm)	PL Peak (nm)	Τ1 (ns)	Τ2 (ns)	Average Lifetime (ns)	QY (%)
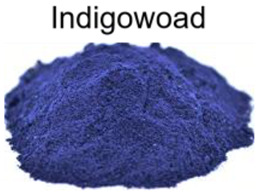	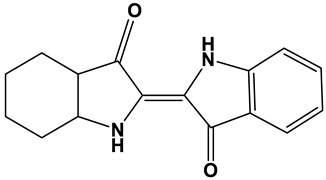	220–450	400	1.6	8.7	3.0	3.8
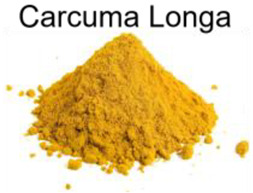	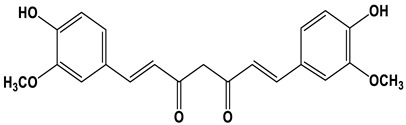	220–450	460	0.3	5.2	3.9	0.8
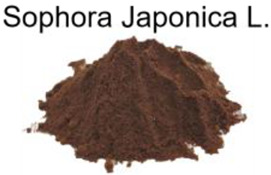	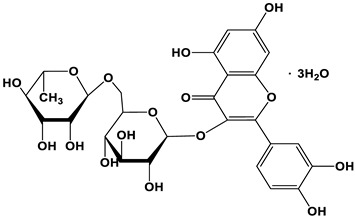	220–450	420	0.9	4.5	1.0	1.0

## Data Availability

The data presented in this study are available on reasonable request from the corresponding author.

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
