# Peer review of "Dye Plants Derived Carbon Dots for Flexible Secure Printing"

_nanomaterials, 2022, doi:10.3390/nano12183168_

Round 1

Reviewer 1 Report

Dye Plants Derived Carbon Dots for Flexible Anti-counterfeiting

Authors have conducted an interesting, novel research project with carbon dots emphasizing the importance of carbon dots in various applications. The research data gathering is admirable. However, the manuscript needs to be critically revised in order to uplift the quality.  

1) Authors have mentioned in the introduction the following statement 'While the current used precursors for C-dots production are still toxic compared to 46 natural biomass." I do not agree with this statement because there are so many carbon dots have been synthesized from pure natural carbon sources and proved to be non-toxic. Thus, I suggest the authors refer few more previously published manuscripts and revise the sentence.

2) I suggest the authors to conduct a toxicity study with cell lines or sea-urchin to prove your carbon dots dyes are non-toxic.

3) The authors mentioned in the introduction that the carbon dots were synthesized via the hydrothermal method. However, in the methodology, they have mentioned it was synthesized through heating in the autoclave. The term "hydrothermal" is used when the synthesis is carried out in a water bath at a particular temperature. Thus, I suggest the authors revise the term hydrothermal into the autoclave.

4) Majority of the carbon dots are water-soluble. Thus, I would like to see if the authors can describe the solubility conditions of these carbon dots dyes. I wonder whether these carbon dots-based prints are washing off when contacted with water.

5) I encourage authors to revise the title of the manuscript, especially the word "anti-counterfeiting." I believe if the authors can direct these carbon dots dyes for particular applications, such as fabric printing or secure printing, the readers' complete attention will be drawn to the manuscript. 

Author Response

Comments to the Author

Authors have conducted an interesting, novel research project with carbon dots emphasizing the importance of carbon dots in various applications. The research data gathering is admirable. However, the manuscript needs to be critically revised in order to uplift the quality. 

We appreciate the time and effort invested by the referee in reviewing our work.

In the following, we included a point by point response to the reviewer’s comments.

1) Authors have mentioned in the introduction the following statement 'While the current used precursors for C-dots production are still toxic compared to 46 natural biomass." I do not agree with this statement because there are so many carbon dots have been synthesized from pure natural carbon sources and proved to be non-toxic. Thus, I suggest the authors refer few more previously published manuscripts and revise the sentence.

Thank you for your comments. We have removed this claim in the revised MS.

2) I suggest the authors to conduct a toxicity study with cell lines or sea-urchin to prove your carbon dots dyes are non-toxic.

Thank you for your comments. It is very interesting to conduct a toxicity study with cell lines or sea-urchin. While due to the limited resources in our lab, it is difficult for us to conduct such experiments for now. We strongly believe that we would like to conduct such measurement in the future and reported any of results in other publication.

3) The authors mentioned in the introduction that the carbon dots were synthesized via the hydrothermal method. However, in the methodology, they have mentioned it was synthesized through heating in the autoclave. The term "hydrothermal" is used when the synthesis is carried out in a water bath at a particular temperature. Thus, I suggest the authors revise the term hydrothermal into the autoclave.

Thank you for your comments. We have modified this expression throughout the MS according to the comment. In fact, we used a heating approach in an autoclave reactor.

4) Majority of the carbon dots are water-soluble. Thus, I would like to see if the authors can describe the solubility conditions of these carbon dots dyes. I wonder whether these carbon dots-based prints are washing off when contacted with water.

We are extremely grateful to you for pointing out this issue. The C-dots we have prepared in this work have good solubility due to the surface negative charge (new fig. S8). Per your suggestion, at current condition, we can partially remove the pattern by washing. While we can improve the water-resistance of the pattern by post-treatment. Such study has been carried out in our lab.

5) I encourage authors to revise the title of the manuscript, especially the word "anti-counterfeiting." I believe if the authors can direct these carbon dots dyes for particular applications, such as fabric printing or secure printing, the readers' complete attention will be drawn to the manuscript.

Thank you for your suggestion. We revised the title as “Dye Plants Derived Carbon Dots for Flexible Secure Printing”.

Reviewer 2 Report

The authors have investigated the use of plant derived dye carbon dots in application of anti-counterfeiting. However, the referee believes more results and data are needed as below.

The reviewer suggests to;

1.       add AFM images along with TEM to validate the spherical claims in x-y-z planes.

2.       Zeta potential measurements needs to be conducted to understand the surface potential properties of these dots.

3.       Along with zeta potential studies, stability studies need to be presented to understand the usability of these dots in such applications. In here carbon dots need to be investigated for physical stability against dot aggregation which is identified very common for such material, as well for optical stability to suit in such application.

Author Response

Comments to the Author

The authors have investigated the use of plant derived dye carbon dots in application of anti-counterfeiting. However, the referee believes more results and data are needed as below.

We appreciate the time and effort invested by the referee in reviewing our work. In the following, we included a point by point response to the reviewer’s comments.

(1) add AFM images along with TEM to validate the spherical claims in x-y-z planes.

Thank you for your comments. We conducted the AFM, while we cannot obtain very clear images. In such case, per your suggestion, we conducted TEM again and obtained clearer TEM image compared to our previous version (New figure 1). If the C-dots have 2D structure, it might be very difficult to obtain the lattice from the HRTEM in such small size, it is highly possible that the C-dots has quasi-spherical shape.

(2) Zeta potential measurements needs to be conducted to understand the surface potential properties of these dots.

Thank you for your comments. We conducted the Zeta potential measurements (Fig. S8) and added the relevance text in the MS.

(3) Along with zeta potential studies, stability studies need to be presented to understand the usability of these dots in such applications. In here carbon dots need to be investigated for physical stability against dot aggregation which is identified very common for such material, as well for optical stability to suit in such application.

Thank you for your comments. After 3-month storage, the PL of the C-dots is still stable, maintaining 80% of their initial PL values. We added this figure in the SI and added also the text in the revised MS.

Author Response

Dear Reviewer:

We appreciate the time and effort invested by the referee in reviewing our work. We feel very sorry for the error in the analysis of the article, we took it very seriously and rechecked the content of the article carefully based on the data and the changes made are marked in red in the article.

We believe that the revised version is suitable for publication in Nanomaterials.

Round 2

Reviewer 1 Report

The authors have systematically revised the manuscript according to the review's suggestions. However, I encourage authors to include some data that prove the water resistance of the carbon dots based on printed patterns by post-treatment. Because, if the prints are washable with water, they will wash off if the person wears carbon dots printed clothes in the rain. So, it is better to prove that even though the carbon dots are water soluble, you can still make them water resist after the printing. 

Author Response

We are extremely grateful to you for pointing out this issue. Due to the good solubility of the ink, the pattern can be removed by washing. We tried to spin a thin layer of PMMA or PE polymer on the both side of the cotton to avoid any contact of the moisture or water. We found that the pattern can maintain mostly in the cotton.

Meanwhile we would like to emphasize that we still can use the pattern without washing, such as label for various types of textiles.  

Reviewer 2 Report

The authors have answered to most of the reviewer’s comments and thus, is suitable for accept.

Author Response

We appreciate the time and effort invested by the referee in reviewing our work.